# The Impact of Dietary Factors on the Sleep of Athletically Trained Populations: A Systematic Review

**DOI:** 10.3390/nu14163271

**Published:** 2022-08-10

**Authors:** Jackson Barnard, Spencer Roberts, Michele Lastella, Brad Aisbett, Dominique Condo

**Affiliations:** 1Centre for Sport Research, Deakin University, Geelong, VIC 3220, Australia; 2Appleton Institute for Behavioural Science, Central Queensland University, Wayville, SA 5034, Australia; 3Institute for Physical Activity and Nutrition (IPAN), School of Exercise and Nutrition Sciences, Deakin University, Geelong, VIC 3220, Australia

**Keywords:** athlete, caffeine, carbohydrates, diet, health, nutrition, protein, sport

## Abstract

Many athletic populations report poor sleep, especially during intensive training and competition periods. Recently, diet has been shown to significantly affect sleep in general populations; however, little is known about the effect diet has on the sleep of athletically trained populations. With sleep critical for optimal recovery and sports performance, this systematic review aimed to evaluate the evidence demonstrating that dietary factors influence the sleep of athletically trained populations. Four electronic databases were searched from inception to May 2022, with primary research articles included if they contained a dietary factor(s), an outcome measure of sleep or sleepiness, and participants could be identified as ‘athletically trained’. Thirty-five studies were included, with 21 studies assessed as positive quality, 13 as neutral, and one as negative. Sleep or sleepiness was measured objectively in 46% of studies (*n* = 16). The review showed that evening (≥5 p.m.) caffeine intakes >2 mg·kg^−1^ body mass decreased sleep duration and sleep efficiency, and increased sleep latency and wake after sleep onset. Evening consumption of high glycaemic index carbohydrates and protein high in tryptophan may reduce sleep latency. Although promising, more research is required before the impact of probiotics, cherry juice, and beetroot juice on the sleep of athletes can be resolved. Athletic populations experiencing sleep difficulties should be screened for caffeine use and trial dietary strategies (e.g., evening consumption of high GI carbohydrates) to improve sleep.

## 1. Introduction

Sleep is fundamental for physiological and psychological health, with adults recommended to obtain seven to nine hours’ sleep per night for optimal wellbeing and functioning [1]. Athletic populations may require additional high quality sleep relative to non-athletes to adequately recover from stressors like training and competition [2,3]. Whilst anecdotal evidence suggests that athletes consider sleep to be the single best recovery strategy [4,5], elite athletes frequently sleep less than seven hours per night [3,6], with up to 65% of athletes classified as poor sleepers through subjective screening tools [7]. For an athlete, poor sleep compromises multiple aspects of recovery and performance, with greater risks of illness and injury; and cognition, pain perception, and sports-specific skill execution (i.e., goal keeping reaction times and tennis serving accuracy) impaired [8,9,10,11,12,13], all of which may diminish sporting success [14,15]. The high prevalence of athletes reporting poor sleep is an area of concern given the prospect that inadequate sleep impairs athletic performance and wellbeing [16]. Therefore, identifying proven, practical strategies to improve the sleep of athletes is warranted.

Sleep hygiene is a strategy that refers to improving behavioural and environmental variables that can affect sleep [17]. Whilst improving sleep hygiene may generally be an effective strategy to acutely improve sleep in athletes [18], recommendations to avoid high-intensity exercise in the evening, maintaining regular sleep/wake times, and optimizing the sleep environment can be compromised through factors outside of an athlete’s control [19,20] Athletes face many barriers to sleep, including intensive training and competition scheduling, travelling across time zones, and pre-competition anxiety [20]. Additionally, strict anti-doping and medication policies may limit an elite athlete’s ability to trial effective sleep aids such as melatonin [19]. With common sleep-promoting recommendations having little translation to the unique athlete environment, athletes and their coaches need to explore alternatives.

Recent reviews link diet to sleep quality and duration [21,22,23,24,25], expanding the essential role that nutrition plays in sports performance and recovery. Common dietary themes associated with improved sleep in non-athletes include meal timing, macronutrient composition, and dietary supplementation [21,22,23,24,25]. For instance, meal timing and nutritional composition are primary cues for peripheral molecular clocks including those in the liver and gut [26,27]. Alterations of these peripheral clocks through changes in meal timing and composition may induce circadian phase shifts, which can result in circadian misalignment [26,28,29]. With sleep coinciding with decreases in core body temperature [30,31], eating close to bedtime can theoretically impair sleep through metabolic disturbances associated with digestion. Further, meal composition can influence the secretion of sleep-regulating hormones such as melatonin [28,32]. Melatonin is primarily involved in conveying light/dark signals to body physiology, playing a major role in the regulation of the sleep/wake cycle [33]. Data suggests that protein and carbohydrate intake likely influence melatonin synthesis indirectly through the tryptophan (TRP) to large neutral amino acid (LNAA) ratio [28,32,34]. Tryptophan is an amino acid that shares the same transport system as other LNAAs to cross the blood–brain-barrier [32]; therefore, the ratio of TRP:LNAAs within blood plasma affects tryptophan availability to the brain and resultant melatonin production [28,32,34]. Increases in the TRP:LNAA ratio may promote TRP availability, melatonin synthesis, and ultimately sleep, which occurs through the intake of carbohydrates or protein sources high in TRP and low in other LNAAs (e.g., dairy) [34,35,36]. Alternatively, an intake of protein sources high in LNAA and low in TRP reduces the TRP:LNAA ratio and may hinder sleep [34,37,38], but this has not been tested extensively in athletes.

Nutritional recommendations for optimizing athlete sleep primarily draw upon literature from the general population [13,16,39]. This may not be appropriate for athletes, however, with a recent review finding macronutrient intake may influence the sleep quality of athletic and non-athletic populations differently [39]. Consumption of high glycaemic index (GI) carbohydrates with an evening meal resulted in more disrupted sleep within state-level basketballers [40], but resulted in shortened sleep latency within a general population [39,41]. Similarly, a high tryptophan protein supplement consumed in the evening improved multiple aspects of sleep quality in healthy males [42], yet had no impact on the sleep of well-trained male cyclists [43]. Though the full extent of athlete literature within this area is still to be established, it may be speculated that differences in macronutrient intake [37,44], nutritional requirements [45], physiological adaptation [46], sleeping behaviours [47,48], and stressors including exercise and competition [20] may have led to discrepancies between the nutrition–sleep relationships demonstrated for non-athletic and athletic populations.

With growing interest in the nutrition–sleep relationship of athletic populations [16,39], it is important that nutritional recommendations are guided by data within athletic populations, as athlete-specific differences may add confounding factors to the nutrition–sleep relationship. Therefore, the aim of this systematic review was to evaluate the evidence for the impact of specific dietary factors including composition, timing, and dietary supplement use on sleep duration and quality within athletically trained populations.

## 2. Materials and Methods

This review follows the Preferred Reporting Items for Systematic Reviews and Meta-Analyses (PRISMA) guidelines, with PROSPERO trial registration number CRD42021264200. The PRISMA checklist is reported in Appendix A.

### 2.1. Search Strategy

Four electronic databases (CINAHL Complete, Embase, MEDLINE Complete, and SPORTDiscus with Full Text) were searched from database inception to May 2022. The following search terms and strategy were used for all databases: (Athlet* OR Sport* OR Player* OR Exercis* OR Active OR Elite OR Trained OR Compet*) AND (Nutri* OR M?cronutri* OR Food OR Diet OR Diets OR Meal* OR Supplement*) AND (Sleep* OR Nap OR Naps OR Napping)). The search was limited only to human studies available in English.

### 2.2. Eligibility Criteria

Included studies met the following criteria: (1) participants were stated as being athletes (e.g., student, sub-elite, elite, and/or professional), competitive sportspeople, trained (e.g., endurance-trained, resistance-trained), or could be identified as ‘athletically trained’ using previously published criteria (see Section 2.2.1); (2) participants within the study were healthy non-injured individuals; (3) the study included a nutrient and/or food source as the independent variable; (4) the study reported at least one subjective and/or objective outcome measure of sleep or sleepiness; (5) the mean age of participants was ≥18 years; (6) the study was an original research study; (7) the study was published in English as a full-text article in a peer-reviewed journal.

Although athletically trained criteria may extend to ‘tactical’ athletes (i.e., law enforcement, military, and rescue personnel that require physical training to optimize occupational performance [49]), these populations were excluded from the review due to difficulties determining the training status and requirements of these population groups.

#### 2.2.1. Trained Population Classification

Classification of ‘athletically trained’ subjects complied with guidelines for sports-science research [50]. These cycling-based guidelines use ‘trained’ terminology when training is completed ≥3 times per week, for ≥5 h per week [50]. Therefore, for this review, these guidelines were extrapolated to consider populations as ‘trained’ if exercising ≥ 3 times per week, for ≥5 h per week. As such, studies detailing their participant’s training frequency and duration at these amounts were deemed eligible for inclusion.

### 2.3. Study Selection and Data Extraction

Articles identified in the systematic search were exported to Covidence software (Melbourne, Australia) following duplicate removal. The title and abstract of each retrieved record were screened by one reviewer (JB). The full text of each relevant screened article was then examined by two reviewers (JB, SR) to determine whether inclusion criteria were met. Any discrepancies were discussed to reach consensus.

Data from eligible studies were extracted by one reviewer (JB) into a spreadsheet designed to record information on the study design, methods, and sample population. The spreadsheet of extracted data was then screened for accuracy by another reviewer (SR). The following variables were extracted from eligible articles: author (year), study design, sample size, participant details (age, sex, sport type, training status), study duration, intervention/control methodology, sleep assessment methodology, results, and conclusions. The mean ± SD for sleep outcomes of interest were recorded (see Section 2.3.1). Studies were then categorized into themes according to the dietary factor investigated.

#### 2.3.1. Sleep Definitions and Outcomes of Interest

This systematic review included studies reporting objective and/or subjective measures of sleep or sleepiness. Table 1 provides the definitions of key sleep-related terms used throughout this review [6].

Sleep quality is a widely used term with no single definition [51]. Subjective sleep quality is defined in relation to feeling rested and restored upon waking, tiredness during the day, and number of awakenings at night [52]. Objectively, good sleep quality as determined by the National Sleep Foundation, is defined in relation to sleep onset latency (≤15 min), number of awakenings greater than five minutes (≤1), wake after sleep onset (≤20 min), and sleep efficiency (≥85%) [51].

### 2.4. Quality Assessment

The methodological quality and risk of bias within each study was independently assessed by two reviewers (JB, SR), using the Academy of Nutrition and Dietetics Quality Criteria Checklist: Primary Research tool [53]. Discrepancies were discussed between the reviewers, with another author (DC) consulted if a consensus could not be reached. The quality criteria checklist contains ten validity questions relating to: (1) research questions; (2) participant selection; (3) study groups’ comparability; (4) participant withdrawal; (5) study blinding; (6) interventions and comparisons; (7) outcome measures; (8) statistical analysis; (9) conclusions; (10) conflicts of interest. Study quality was classified as positive (+) if question 2, 3, 6, and 7, as well as one other question were scored with a ‘yes’; neutral (ø) if any of the questions 2, 3, 6, and 7 did not receive a ‘yes’; or negative (−) if six or more questions were answered with a ‘no’. 

## 3. Results

A total of 10,211 articles were identified through four electronic databases and citation searching. There were 2991 duplicates removed, and of the 7220 records remaining, 7101 articles were removed based on title and abstract as they did not include trained populations, sleep outcomes, or a dietary factor. After assessing 119 full-text articles for eligibility, 35 articles were included for review. The PRISMA flow chart for the systematic review is presented in Figure 1.

### 3.1. Study Characteristics

Thirty-five studies were included in this systematic review (Table 2, Table 3 and Table 4) [37,40,43,54,55,56,57,58,59,60,61,62,63,64,65,66,67,68,69,70,71,72,73,74,75,76,77,78,79,80,81,82,83,84,85]. These studies were categorized into main themes based on the dietary factor(s) examined; *macronutrients, micronutrients* and *energy*, *dietary supplements*, and *dietary patterns*. All studies were published between 2010 and 2022, with 83% (*n* = 29) published between 2018 and 2022 [37,40,43,54,57,58,59,60,61,62,64,65,66,69,70,71,72,73,74,75,77,78,79,80,81,82,83,84,85]. Females were recruited in 18 studies (51%) [43,54,58,60,61,62,67,68,71,76,78,79,80,81,82,83,84,85], with only 23% of studies (*n* = 8) recruiting females exclusively [54,58,61,62,68,76,82,85], compared to 49% for males [37,40,55,56,57,59,63,64,65,66,69,70,72,73,74,75,77]. Of the 35 studies, 21 were randomized control trials [40,43,55,56,57,58,59,60,61,63,65,67,68,69,70,72,73,74,75,76,85], four were prospective cohort studies [37,54,64,66], and ten were cross-sectional survey designs [62,71,77,78,79,80,81,82,83,84]. For the 21 randomized control trials, a total of 314 participants were recruited, with only 26% (*n* = 82) of these being female. The mean age of participants included in the review ranged from 18.0 to 39.5 years. The training status of participants varied, with recruited populations described as *elite/professional* (*n* = 18) [37,43,54,59,62,64,66,67,69,70,71,72,73,75,77,80,81,83], *sub-elite* (*n* = 1) [71], *national/international* level (*n* = 4) [65,68,79,82], *state-level* (*n* = 1) [40], *well/highly trained* (*n* = 4) [43,56,63,76], *trained* (*n* = 4) [55,60,61,85], *recreationally-trained* (*n* = 2) [57,68], and *active* (*n* = 1) [58], with three studies not reporting training status [74,78]. Studies not reporting training status were included in the review as participants were described as athletes [74,78,84], and the *active* population were completing moderate-vigorous activity >4 days per week [58]. Participants were recruited from numerous sports, including Australian football (*n* = 3) [37,54,59], basketball (*n* = 2) [40,69], cycling (*n* = 4) [43,56,60,63], multiple individual and team sports (*n* = 9) [61,68,71,78,80,81,82,83,84], Paralympic sports (*n* = 1) [79], rhythmic gymnastics (*n* = 1) [62], rugby (*n* = 6), running (*n* = 2) [65,76], soccer (*n* = 1) [73], swimming (*n* = 1) [67], and triathlon (*n* = 3) [55,63,76], with four studies not specifying their participants involvement in sports [57,58,74,85]. Studies were mostly conducted in Australia (*n* = 9) [37,54,59,61,63,64,66,71,72], followed by Japan (*n* = 5) [78,79,81,82,83], Spain (*n* = 3) [65,69,73], United Kingdom (*n* = 3) [56,70,77], and New Zealand (*n* = 3) [67,68,75].

### 3.2. Evidence Quality and Data Collection Methods

Using the Academy of Nutrition and Dietetics Quality Criteria Checklist: Primary Research tool (Appendix A), 21 of the 35 studies were rated as positive primary research studies [37,40,54,55,56,57,58,59,60,62,63,64,65,66,70,73,79,80,82,84,85]. Thirteen studies were rated as neutral, as they did not satisfy all required criteria (criterion 2, 3, 6, and 7) [43,61,67,68,69,71,72,75,76,77,78,81,83]. One study was rated as negative, as this studied satisfied only three of the 10 validity questions [74]. Although the majority of studies were classed as positive based on the criteria [53], many studies provided inadequate information surrounding withdrawal and attrition rates of their participants [37,40,43,54,55,57,59,65,66,67,68,69,70,71,72,74,76], and seven studies measuring only subjective sleep outcomes did not use validated sleep tools [67,69,71,72,76,77,83], six studies provided limited information or used inappropriate statistical methods for the study design [67,71,74,75,76,82], five studies had concerns around blinding of participants and/or researchers [40,55,63,70,74], two studies did not discuss the limitations of their studies [71,74], and two did not report randomization techniques used for participant condition assignment [74,75]. Most neutral rated studies satisfied the majority of criteria outside of criterion two (assessing selection bias).

Sixteen studies objectively recorded sleep [37,40,43,54,55,56,57,58,59,60,61,63,64,65,66,73], with 19 providing only subjective measures of sleep [62,67,68,69,70,71,72,74,75,76,77,78,79,80,81,82,83,84,85]. Three studies recorded sleep via polysomnography (PSG) [57,61,63], and 13 through actigraphy [37,40,43,54,55,56,58,59,60,64,65,66,73]. Of those that used actigraphy, 8% (*n* = 1) used a high sleep-wake threshold (>80 activity counts) [65], 23% (*n* = 3) a medium threshold (>40 activity counts) [37,54,59], 8% (*n* = 1) used the Cole–Kripke algorithm [43], and 15% (*n* = 2) used the ‘Readiband’ device-specific software [58,64]. The sleep-wake threshold/algorithm was not reported by 46% (*n* = 6) of studies using actigraphy [40,55,56,60,66,73]. The most common subjective sleep data collection tools included the Pittsburgh Sleep Quality Index (*n* = 8) [62,74,78,79,80,81,82,85], Epworth Sleepiness Scale (*n* = 4) [40,61,62,81], sleep quality rated on a 1–5 scale (*n* = 4) [65,70,72,75], and the Karolinska Sleepiness Scale (*n* = 3) [43,60,80].

### 3.3. Qualitative Synthesis

#### 3.3.1. Macronutrients and Energy

##### Carbohydrates

Five studies assessed the influence of carbohydrates on the sleep of athletic populations (Table 2) [37,40,55,56,57]. Two studies investigated the influence of GI on sleep [40,57]. In recreationally-trained individuals, a high GI meal consumed immediately post-evening exercise increased PSG derived total sleep time (+1 h) and sleep efficiency (+8.1%), and reduced sleep latency (−18.9 min) and WASO (−32.9 min) compared to a low GI meal [57]. Contrasting this, the sleep of state-level basketballers was unaffected by the GI of the evening meal and snack they consumed [40].

Following 10 consecutive days of diet and sleep tracking via actigraphy, the sleep of elite male Australian footballers was associated with carbohydrate timing (Table 2) [37]. Total daily carbohydrate intake (mean 3.4 ± 1.4 g∙kg∙d^−1^) was not associated with sleep; however, for every 1 g∙kg^−1^ increase in evening (post 6 pm) sugar intake, there was an associated reduction of TST by five minutes (*p =* 0.027), increase in sleep efficiency of 0.2% (*p* = 0.021), and a decrease in WASO by one minute (*p* = 0.005). In elite female Australian footballers, however, total daily carbohydrate intake (mean 3.3 ± 1.3 g∙kg∙d^−1^) was associated with impaired sleep [54]. For every 1 g∙kg^−1^ increase in total daily carbohydrate intake, there was an associated reduction in sleep efficiency by 0.6% (*p* = 0.007), and an increase in WASO by 3.6 min (*p* = 0.007). Further, in trained triathletes, a three-week sleep-low carbohydrate intervention slightly reduced sleep efficiency by 1.1% (*p* < 0.05) compared to a control diet, with no other sleep metrics affected [55]. This sleep-low strategy referred to carbohydrate availability, with the sleep-low group having carbohydrate intake restricted throughout exercise and dinner, resulting in low carbohydrate availability at rest (Table 2). Similarly, a 10-day high carbohydrate intervention (9.9 ± 1.5 g∙kg∙d^−1^ vs. 7.4 ± 1.6 g∙kg∙d^−1^) within highly-trained male cyclists resulted in minimal influences on sleep, with total sleep time slightly decreased (0.3 h) compared to control (*p* = 0.03) [56]. This 10-day period involved a 1.5-fold increase in typical training volume, which was independently associated with an increase of six wake bouts (*p* = 0.03) and a five percent reduction in sleep efficiency (*p* < 0.05).

##### Protein

One three-armed crossover trial investigated the influence of protein type (cottage cheese, casein, and placebo) on the sleep of 10 active females (Table 2) [58]. The cottage cheese and casein contained 30 g of protein and 10 g carbohydrates, with the placebo being non-nutritive. These foods were consumed 30–60 min before bed for one night, and did not influence actigraphy derived total sleep time, sleep latency, or sleep efficiency across intervention groups. These findings were similar to a 10-day prospective cohort study of elite female Australian footballers, whereby a mean intake of 1.8 ± 0.6 g∙kg∙d^−1^ body weight of protein was not associated with sleep. This differed to a cohort study of elite male Australian footballers, with increases in total daily and evening protein intake associated with multiple sleep metrics tracked through actigraphy. Every 1-g∙kg^−1^ increase in daily protein intake (mean 2.2 ± 0.8 g∙kg∙d^−1^) related to a decreased sleep efficiency of 0.7% (*p* = 0.006) and an increased WASO of four minutes (*p* = 0.013), whereas each 1 g∙kg^−1^ increase in evening protein intake was associated with a reduction of sleep latency by two minutes (*p* = 0.013) [37].

##### Whey Protein and Alpha-Lactalbumin

Four studies have objectively measured the sleep of athletically trained populations post-evening consumption of protein sources rich in the amino acid tryptophan (Table 2) [43,59,60,61]. One crossover study that supplemented elite male Australian footballers with 55 g of whey protein (containing 1-g TRP) or an isocaloric placebo on a training and non-training day observed no effect on sleep [59].

Three studies investigated the impact of α-lactalbumin supplementation [43,60,61]—the richest food source of TRP [86]—on sleep. Most sleep metrics measured across the three studies were unaffected following 40–60 g α-lactalbumin (containing 2–3 g TRP) supplemented two hours before bed for 2–3 nights. Total sleep time ranged across these studies from 6.9 to 7.7 h, sleep latency from 2 to 25 min, sleep efficiency between 87.2 and 90.5%, and WASO between 38 and 48 min across all experimental groups [43,60,61]. The two studies that observed no changes in sleep were measured via actigraphy [43,60], whilst a significant increase in N-REM stage 2 sleep was observed in trained females following PSG measurement [61]. Additionally, these females completed a simulated evening competition (18:30–20:00) prior to supplementation and sleep.

##### Fats

Two prospective cohort studies of elite Australian footballers investigated the influence of macronutrients on sleep across a 10 day period in the preseason [37,54]. In male Australian footballers, dietary fat intake (mean 1.7 ± 0.7 g∙kg∙d^−1^) was not associated with sleep [37]. In female Australian footballers, however, increases in saturated fat intake was associated with reductions in actigraphy measured sleep latency [54]. For every 1-g increase in saturated fat intake, there was an associated reduction of sleep latency by 0.27 min (*p* = 0.030).

##### Energy

In a study of 67 international level rhythmic gymnasts, 82% were found to consume <2000 kCal·d^−1^ (Table 2) [62]. There were no differences identified between the subjective sleep quality, duration, or daytime sleepiness in gymnasts consuming either <2000 kCal·d^−1^ or ≥2000 kCal·d^−1^, as assessed through the PSQI and Epworth Sleepiness Scale [62]. One female cohort study of Australian footballers similarly found no association between daily energy intake (mean 9347 ± 2905 kJ (~2234 kCal)) and sleep, whereas two studies of male athletes presented contrasting findings. A prospective study of 36 elite Australian male footballers observed that increases in daily energy intake (mean 14000 ± 4000 kJ (~3346 kCal)) were significantly associated with actigraphy-derived measures of sleep latency and WASO [37]. For every 1 MJ increase in daily energy intake, there was an associated increase in WASO of three minutes (*p* = 0.032), with every additional 1 MJ consumed in the evening associated with an increase in sleep latency by five minutes (*p* = 0.011). Likewise, in state-level basketballers, increases in energy intake were associated with reduced sleep durations (*p* < 0.05) [40].

#### 3.3.2. Micronutrients

One cohort study of elite female Australian footballers investigated the impact of habitual micronutrient intake within the preseason on sleep (Table 2) [54]. Sleep was measured via actigraphy, with associations between micronutrients and sleep reported per unit increase in micronutrient consumption. A longer TST was associated with higher iron intake (per 1 mg, 0.55 min); an increased sleep efficiency was associated with higher iron (per 1 mg, 0.05%), vitamin B12 (per 1 μg, 0.4%), and zinc intake (per 1 mg, 0.23%); a reduction in sleep latency was associated with higher calcium (1 mg, 0.005 min) and magnesium intake (per 1 mg, 0.02 min); a reduction in WASO was associated with higher vitamin B12 intake (per 1 μg, 1.72 min); whereas a reduction in TST was associated with higher sodium intake (per 1 mg, 0.012 min); and a reduction in sleep efficiency was associated with higher vitamin E intake (per 1 mg, 0.08%).

#### 3.3.3. Dietary Supplementation

##### Caffeine

Eight studies investigated the impact of caffeine on sleep [63,64,65,66,67,68,69,84], with all studies observing a negative effect on either sleep duration and/or quality (Table 3). Most studies had participants consume caffeine before evening exercise (e.g., ≥5 p.m.), with intake ranging from 45–75 min pre-session [63,64,65,66,67,68,69]. One study supplemented participants with caffeine prior to a morning swimming trial (09:00–11:30 am) [67]. Four studies objectively measured sleep [63,64,65,66], with evening caffeine intake ranging from ~2 to 6 mg∙kg^−1^ body mass (Table 3). Three studies observed reductions in total sleep time (−1.2–2.8 h) [63,64,66] and sleep efficiency (−5.8–15.4%) [63,64,65], and an increased sleep latency (+7.8–40.9 min) [63,64,66]. Wake after sleep onset was increased in two studies (+22.9–33.2 min) [63,65], with an increase of 5.3 awakenings observed in another study following caffeine conditions [65]. Compared to placebo, caffeine conditions negatively affected subjective measures of sleep including total sleep time [67], sleep latency [65,67,68], sleep quality [65,68], and prevalence of insomnia [69] across multiple studies. In endurance athletes, consuming ≤1.5 cups of caffeinated beverages per day was associated with a significantly better sleep quality and a lower sleep difficulty (*p* < 0.05) as assessed via the Athlete Sleep Screening Questionnaire (ASSQ) [84].

##### Cherry Juice

Using subjective measures, two studies explored the impact of cherry juice on the sleep of athletes [70,71]. In professional rugby league athletes, supplementation of 2 × 30 mL Montmorency cherry juice (morning and night) for eight days had no influence on subjective sleep quality, as measured via a 1–5 Likert Scale (Table 3) [70]. In contrast, a survey of 80 ≥ sub-elite athletes revealed that 14% of the athletes who had previously or are currently supplementing a non-specified amount of tart cherry juice reported improved sleep (Table 3) [71]. The general questionnaire used did not specify the type of sleep improvement experienced, so it is unknown whether 14% of these athletes experienced improvements in sleep quality or duration.

##### Pre and Probiotics

Two randomized control trials supplemented professional rugby union or soccer players with probiotics and measured sleep outcomes for a period of 30 days to 17 weeks (Table 3) [72,73]. Following 30 days supplementation of a 300-mg mixture of probiotic (*Bifidobacterium lactis*, *Bifidobacterium longum,* and *Lactobacillus rhamnosus*) and prebiotic *(fructooligosaccharides)* strains, actigraphy-derived measures of sleep latency and sleep efficiency were improved in Spanish soccer players by 0.5 min and 3.3%, respectively [73]. Further, bi-daily supplementation of a proprietary probiotic blend (*Lactobacillus*, *Bifidobacterium,* and *Streptococcus*), improved the subjective sleep quality of professional Australian rugby union players compared to placebo [72]. Athletes were studied for 17 weeks in-season, with eight weeks domestic competition and nine weeks of international competition. The probiotic group were supplemented with an additional two SBFloractiv™ (*Saccharomyces boulardii*) probiotic capsules daily during international travel to limit traveller’s diarrhoea, which the placebo group did not receive.

##### Other Dietary Supplements

Male athletes supplemented with 100 mL of beetroot juice (300 mg nitrates) two hours pre-exercise (exercise timing not reported) for seven days, significantly improved subjective sleep quality compared to placebo (Table 3) [74]. The PSQI global score of the athletes in the beetroot juice group was reduced by more than half from baseline (14.5–5.8) [74]. This study was reviewed as being negative quality, given the lack of information around participants, interventions (i.e., exercise type and timing, dietary controls) and reporting of data. In professional rugby union males, the addition of 1546 mg omega-3 to protein shakes consumed after a morning and afternoon exercise session had no effect on subjective sleep quality compared to placebo [75]. Similarly, acute supplementation of chocolate milk (12 g protein, 30 g carbohydrates) in trained female runners and triathletes did not influence subjective total sleep time compared to placebo [76]. Lastly, a cross-sectional survey conducted with 517 elite male rugby players displayed that 41% of athletes that were currently or had previously supplemented undefined amounts of cannabidiol (CBD) experienced improved sleep [77]. As no defined CBD amount was included, the dose required to elicit a positive influence on sleep in athletically trained populations remains unknown.

#### 3.3.4. Dietary Patterns

##### Meal Timing and Patterns

Four cross-sectional survey-based studies and one prospective cohort study explored the associations between meal timing on sleep duration and/or quality (Table 4) [78,79,80,81]. These four cross-sectional studies measured sleep quality via the PSQI, and involved populations including student athletes [78], visually-impaired athletes [79], and elite athletes [80,81]. Two studies of student and Paralympic athletes observed no influence of meal timing on subjective sleep quality [78,79]. In contrast, regression analysis displayed an association with eating breakfast every morning and sleep quality (*p* < 0.01) in elite athletes [81], and eating a heavy meal within three hours of bed related to a subjective increase in total sleep time (*p* < 0.05) and WASO (*p* < 0.05) in youth athletes [80]. This contrasts the findings of Falkenberg et al. [37], whereby every additional hour between the main evening meal and bedtime was associated with a decrease in sleep duration by eight minutes (*p* = 0.042), and a decrease in WASO by two minutes (*p* = 0.015) in Australian footballers [37]. Further, an eight-week three-armed randomized control trial observed no significant changes in the PSQI global score of resistance trained females following a control diet or a time-restricted feeding protocol (all calories consumed between 12:00 h and 20:00 h) with and without 3 g·d^−1^ β-hydroxy β-methylbutyrate supplementation [85].

##### Total Diet

A survey-based study of female college athletes investigated the relationship between food and nutrient intake on sleep quality (Table 4) [82]. Using PSQI global scores, athletes were categorized as either being no-risk (score < 5.5) or at-risk for sleep disorders (score ≥ 5.5). As measured via the Food Intake Frequency Questionnaire (Japanese version), there was no significant differences in any nutrient or food group intake between the no-risk or at-risk groups, except for a higher bean intake in the at-risk for sleep disorder group (*p* = 0.034). This differed to a cross-sectional study of endurance athletes, whereby there was no significant influence of wholegrain, fruit, and vegetable consumption on sleep difficulty.

##### Dairy Consumption

In elite female athletes, a medium to high frequency of milk consumption (>3 days per week) was associated with a lower risk of reduced subjective sleep quality (*OR* = 0.38, *p* < 0.001) [83]. Sleep quality was rated on a 1–3 scale, with participant’s intake of dairy and milk categorized as either low, medium, or high. There were no associations between the frequency of milk or dairy consumption on male athlete sleep quality, or total sleep time for either sex (Table 4). Similarly in endurance athletes, dairy consumption was not related to sleep difficulty or quality as assessed via the ASSQ [84].

## 4. Discussion

This is the first systematic review to evaluate the available evidence on the impact of multiple dietary factors on the sleep of athletically trained populations. Consistent with literature in non-athlete populations, results from the 35 studies examined suggests that dietary factors may influence the sleep of athletically trained populations. The main findings from this systematic review were that: macronutrient timing (i.e., evening) and composition (i.e., carbohydrate GI) appear to have a stronger influence on sleep than total daily macronutrient consumption; caffeine consumed before evening exercise negatively impacts sleep; and, sustained probiotic supplementation may promote sleep. These findings can help formulate sports nutrition guidelines to promote sleep in athletically trained populations.

The impact of protein on sleep varied between sexes. In elite male Australian footballers, high protein intake was associated with a lower sleep efficiency and an increased WASO [37]; however, there was no effect of protein on the sleep of their elite female counterparts [54]. Though the effect of protein on sleep efficiency and WASO in the male footballers was small and unlikely meaningful, the discrepancy observed between these studies may be explained by the higher protein intakes of the male footballers (2.2 g∙kg∙d^−1^ vs. 1.8 g∙kg∙d^−1^), or by the type and timing of the protein consumed. Amino acid composition of meals were not assessed in either study; however, the male footballers frequently reported consumption of whey protein in the evening, with evening protein intake associated with reductions in sleep latency. Four studies within this review investigated the relationship between sleep and evening protein supplementation rich in tryptophan [43,59,60,61]—a soporific amino acid that is a precursor to melatonin production [28,32]. Alpha-lactalbumin is the second most predominant protein in whey, containing the highest tryptophan content of food protein sources [36,86]. One study that supplemented male Australian footballers with 55 g of whey protein (containing 1 g tryptophan) in the evening showed no effect on sleep [37]. This same tryptophan content is contained within ~20 g of pure α-lactalbumin [86], meaning the additional 35 g of whey protein required to reach this content may have contributed excessive amounts of LNAA in order to favour TRP availability to the brain. Though it is well established that supplementation of α-lactalbumin increases the TRP:LNAA ratio [36,87,88,89], two studies examining well-trained male cyclists observed no beneficial effect of α-lactalbumin on sleep [43,60]. However, the average sleep efficiency across these two studies was >87% [43,60], and sleep latency was <3 min in one study [60], and previous research suggests that the sedative effects of TRP-based supplementation were less beneficial in populations without sleep complaints [32]. In contrast, α-lactalbumin supplementation improved the sleep of trained females [61], where sleep was measured following evening competition—a typically challenging sleep period for athletes [20]. Although an increase in N-REM 2 sleep was observed in the trained female study, a mixed meal (>24 g protein) was consumed alongside the 40-g α-lactalbumin supplement [61], which likely increased LNAA intake and potentially confounded the effects of the supplement on the TRP:LNAA ratio [38]. This same concept may explain the negative influence total daily protein intake had on sleep, as most protein sources are high in LNAAs and low in TRP [32,34]. Further research into the effects of α-lactalbumin on the sleep of athletically trained populations should aim to recruit individuals with sleep difficulties, and isolate supplementation from mixed meals to limit confounding effects on the TRP:LNAA ratio. With protein requirements increased in athletic populations, further investigation of practical dietary strategies that reduce the impact of high protein intake on athlete sleep (i.e., manipulating protein type and timing) is necessary.

The timing and type of carbohydrates appears influential on sleep. The evening intake of high GI carbohydrates was associated with improvements in sleep latency, WASO, and sleep efficiency in two studies of athletically trained populations [37,57]. In a double-randomized crossover study of recreationally trained males, consumption of a high GI meal (containing 2 g·kg^−1^ carbohydrates) immediately post evening sprint exercises increased total sleep time and sleep efficiency, and decreased WASO and sleep latency compared to a low GI meal as measured by polysomnography [57]. In contrast, within state-level basketballers, no influence on sleep was observed following evening consumption of high GI or low GI carbohydrate-based meals [40]. In addition to exercise being in the morning opposed to evening, this study of basketballers provided meals high in protein (64–81 g), which may have impacted the TRP:LNAA ratio [34,38]. Carbohydrates promote TRP availability to the brain through the action of insulin [34,35], but when included as part of a mixed meal containing high amounts of LNAAs and low TRP, the TRP:LNAA ratio is reduced [38]. Increases in carbohydrate intake were associated with an increased WASO and decreased sleep efficiency in female Australian footballers; however, carbohydrate type was not assessed [54]. Previous research in a non-athletic female population does, however, indicate that high GI diets may increase insomnia risk [90]. With many athletes recommended to adopt high carbohydrate diets to maximize performance and recovery [91], it remains unknown whether there is a trade-off between daily carbohydrate intake, performance, and sleep. This displays the need for more research to determine the impact that carbohydrate type, timing, and total daily amounts have on the sleep of athletic populations, especially for females.

Caffeine inhibits athlete sleep, with doses ≤6 mg∙kg^−1^ body mass affecting numerous sleep measures [63,64,65,66,67,68,69]. The influence of caffeine on sleep is well-established [92]; however, the evidence-base within athletic populations has yet to be consolidated. To achieve the maximal ergogenic benefits of caffeine, it is recommended that caffeine be consumed at doses of 3–6 mg∙kg^−1^ body mass 60 min prior to exercise [93]. Caffeine acutely enhances multiple aspects of performance, with increases in vigilance and awareness resulting from caffeine’s inhibitory action on adenosine receptors within the brain [94]. This sleep-inhibiting action remains after exercise, with caffeine consumption prior to evening exercise resulting in more disrupted sleep [63,64,65,66,68,69]. Consuming caffeine at doses of 5 g∙kg^−1^ of body mass may impair sleep quantity and quality for up to seven hours post consumption in athletes [67]. This aligns with previous data [95], whereby significant sleep disturbances have been observed in healthy adults consuming 400 mg of caffeine six hours prior to bedtime [95]. According to pharmacological data, the average half-life of caffeine is 4–6 h [96], meaning the continued stimulatory effect of caffeine on sleep is dependent on dosage and timing. This is highlighted by the habitual game day use of caffeine (~2 mg∙kg^−1^) in Super Rugby athletes prior to evening competition, whereby caffeine consumed ~50 min before the match resulted in a decreased sleep duration and sleep efficiency, and greater difficulty falling asleep [64]. Knowledge that caffeine impairs athlete sleep for a considerable time post-ingestion allows athletes and practitioners to determine whether the benefits of caffeine on performance outweighs likely disrupted and poorer quality sleep following evening competition and training. Given the large number of athletes supplementing caffeine before evening training and competition, additional next day sleep opportunities should be provided following night-time caffeine supplementation to compensate for likely poorer sleep (i.e., no early morning recovery sessions, adjusting travel schedules to allow for more sleep, etc.).

Chronic probiotic supplementation may improve both objective and subjective sleep measures [72,73]. In professional soccer and rugby players, probiotic supplementation resulted in improvements of multiple sleep metrics [72,73]. Though results are promising, there are limitations to both studies. In soccer players supplemented with a mixture of probiotic strains for 30 days, sleep latency values were under two minutes [73]. This may indicate that athletes either had a significant accumulation of sleep debt [97], or that sleep latency values were underreported via actigraphy, a sleep tool with a potential small bias of ~9.5 min when measuring sleep latency [98]. Actigraphy may, therefore, not be sensitive enough to detect meaningful changes in sleep latency within athletes without sleep difficulties. Further, within the 17-week probiotic study of professional rugby union players, an additional bi-daily probiotic was supplemented only to the treatment group during international periods of the study (nine weeks) to prevent traveller’s diarrhea [72]. The additional travelling probiotic may have confounded the relationship between the experimental probiotic and sleep [72]. Nevertheless, probiotic supplementation is seen to improve subjective sleep quality in non-athletic populations [99], including healthy females [100], and individuals with chronic fatigue [101], chronic pain [102], and stress [103]. Proposed mechanisms by which probiotics may elicit sleep benefits for athletes is through reducing muscle soreness via anti-inflammatory actions [72,104], and through their role in the synthesis of neuropeptides involved in the sleep/wake cycle, including melatonin [105]. Although the preliminary research of probiotics and athlete’s sleep appears hopeful, further studies using more comprehensive means of objective sleep assessment (i.e., polysomnography) and stricter controls of confounding factors (e.g., dietary standardization, travel probiotics) are required to determine the true effect of different probiotic strains on sleep metrics including sleep latency.

Although evidence examining the effect of dietary supplementation on sleep was limited, some dietary supplements appeared to be sleep promoting and, thus, warrant further investigation. The effect of cherry juice on athlete sleep had equivocal findings [70,71], with 14% of rugby players surveyed perceiving non-specified sleep benefits following cherry juice supplementation [71]. Cherry juice provides an exogenous dietary source of melatonin, which has previously benefited the sleep of individuals with or without insomnia in the general population [106,107]. Further, supplementation of 100 mL beetroot juice improved markers of subjective sleep quality in one study of male athletes [74]. The most plausible mechanism of action is that the dietary nitrates contained within beetroot juice increase nitric oxide, a biological messenger involved in the modulation of sleep/wake states [108]. However, this study lacked detail on the population group, methodology, and overall findings, which casts doubt on the repeatability of the findings. With the popularity of beetroot juice use as an ergogenic aid increasing among athletes [109], further high-quality research is necessary to determine the relationship with sleep. Lastly, cannabidiol, a substance no longer prohibited by the World Anti-Doping Agency [110], may promote sleep. In one cohort of rugby players, 41% that had trialled CBD subjectively reported sleep improvements [77]. Although, a wide variety of supplemented CBD doses were reported (400 to 3000 mg), limiting the conclusions that can be drawn between the CBD–sleep relationship in athletic populations. In a large clinical population, 92% of patients that had trialled CBD found it helpful for their sleep [111]. Though CBD supplementation has seen improvements to the sleep of athletes and non-athletes, these studies have relied upon subjective measures of sleep.

## 5. Limitations

Though most studies included in this systematic review were regarded as of neutral or positive quality, many studies provided limited information on dietary controls and withdrawal/attrition rates within their studies. Many studies also did not implement validated subjective sleep tools to determine sleep outcomes, limiting the reliability of results across articles. Less than two-thirds of studies included in this review were RCTs (61%), limiting the ability to determine causality between sleep and certain dietary factors. Females were underrepresented within the literature, with only 20% of participants within the included RCTs being female. As the inclusion of studies within this review relied upon training information and descriptors of participants included by study authors, insufficient information provided within articles may have consequently limited study inclusion. Future studies should, therefore, implement interventional study designs using validated sleep measures, recruit more female participants, limit potential confounders to the nutrition–sleep relationship (e.g., dietary standardization), and report greater detail around participant withdrawal data.

### Practical Applications


Caffeine consumption (>2 mg∙kg^−1^ body mass) prior to evening competition impairs total sleep time, sleep latency, sleep efficiency, and wake after sleep onset.Evening consumption of protein sources high in tryptophan may help promote and maintain the sleep of athletic populations, especially during times of typically disturbed sleep (i.e., after competition).Consumption of high GI carbohydrates immediately after evening exercise may promote sleep.


## 6. Conclusions

This review supports the assertion that an athlete’s diet can impact their sleep. Dietary factors may act as a practical tool to promote or inhibit the sleep of athletically trained populations. Multiple sleep promoting factors were reviewed, including protein sources high in tryptophan, high GI carbohydrate consumption in the evening, and sustained probiotic supplementation. Caffeine was a sleep-inhibiting agent that appeared to negatively affect athlete sleep several hours after consumption. There were few dietary supplements that may promote sleep such as cherry juice and beetroot juice but require further investigation. With the evidence-base displaying that the sleep of athletically trained populations is significantly influenced by diet, practitioners and athletes should strategically manipulate macronutrient timing, type, dietary supplementation, and caffeine intake to optimize sleep and performance outcomes.

## Figures and Tables

**Figure 1 nutrients-14-03271-f001:**
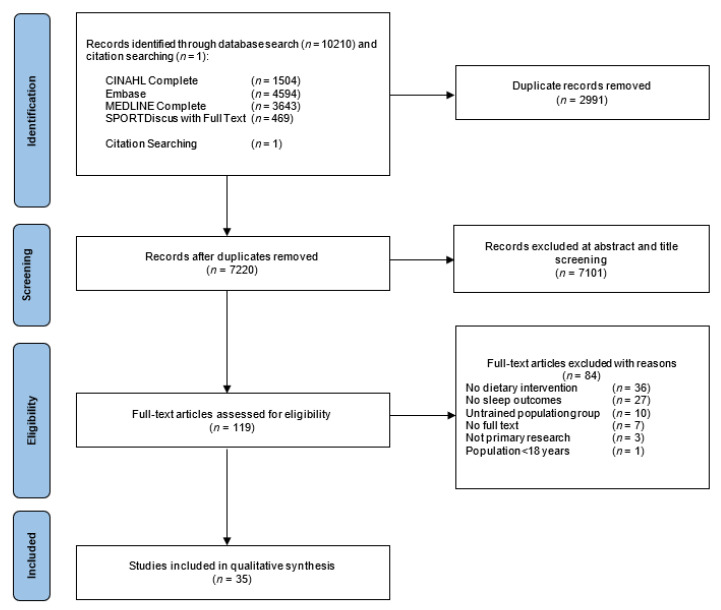
PRISMA flow chart for the selection of included studies.

**Table 1 nutrients-14-03271-t001:** Sleep related definitions [6].

Term	Definition
**Total sleep time (TST)**	The amount of sleep obtained during a sleep period.
**Sleep efficiency (SE)**	The percentage of time in bed that was spent asleep.
**Sleep onset latency (SOL)**	The period of time between bedtime and sleep onset.
**Wake after sleep onset (WASO)**	The amount of time spent awake after sleep has been initiated.
**Sleep stage duration**	The percentage of total sleep time spent in N-REM stage 1, 2, 3, and REM.
**Subjective sleepiness**	The participants’ self-rating of sleepiness, typically ranging from extremely alert to very sleepy.
**Subjective sleep quality**	The participants’ self-rating of sleep quality, typically reported on a Likert-type scale.

**Table 2 nutrients-14-03271-t002:** Studies investigating the influence of *macronutrients, micronutrients,* and *energy* on the sleep of athletically trained populations.

Author(s)	Country	Study Type	Sample Size (m/f)	Age (y)	Sport (Training Status)	Days of Sleep Measurement	Dietary Intervention/Factor		Sleep Tool(s)	Main Outcomes	Study Quality
							Dietary Factor(s)	Timing			
**Carbohydrates**											
Louis et al. [55]	France	RCT	21 (21/0)	31.0 ± 4.7	Triathletes (Trained)	21 (+21 baseline)	All participants consumed 6 g/kg CHO per day **Sleep low:** No CHO intake during exercise sessions and no CHO at dinner **Control:** CHO intake maintained throughout the day and throughout exercise sessions	**Sleep low:** CHO consumed between 08:15–16:00 **Control:** CHO consumed throughout entire day	**Objective:** Actigraphy	Sleep low condition ↓ sleep efficiency compared to control (*p* < 0.05)	+
Killer et al. [56]	United Kingdom	CO (RCT)	13 (13/0)	25.0 ± 5.8	Cyclists (Highly trained)	18	Consumed either a high CHO or isocaloric control nutritional beverage before, during, and after each training session (CHO = ~128 g vs. 33 g)	Before, during, and immediately after exercise (exercise time NR)	**Objective:** Actigraphy	↑TST following control beverage (*p* = 0.03) No significant difference in sleep latency, sleep efficiency, and WASO	+
Vlahoyiannis et al. [57]	Cyprus	CO (RCT)	10 (10/0)	23.2 ± 1.8	NR (Recreationally trained)	2	Receive either a high GI meal or an isocaloric low GI meal after an exercise session	Immediately post-exercise (~2 h pre-bed)	**Objective:** PSG	High GI condition ↑ TST (*p* = 0.019) and sleep efficiency (*p* = 0.049), and ↓ sleep latency (*p* = 0.026) and WASO (*p* = 0.034) compared to low GI	+
Daniel et al. [40]	Brazil	CO (RCT)	9 (9/0)	18.0 ± 0.7	Basketball (State-level)	2	Consume either a high GI dinner and evening snack, or an isocaloric low GI dinner and evening snack	Dinner + evening snack timing NR	**Objective:** Actigraphy	No difference in sleep measures between High GI and low GI conditions	+
Falkenberg et al. [37]	Australia	PC	36 (36/0)	23.5 ± 3.9	Australian football (Elite)	10	Habitual carbohydrate intake and timing	N/A	**Objective:** Actigraphy	Increases in evening (>6 pm) sugar intake associated with ↑ sleep efficiency (*p* = 0.021), and ↓ TST (*p =* 0.027) and WASO (*p* = 0.005)	+
Condo et al. [54]	Australia	PC	32 (0/32)	25.0 ± 4.0	Australian football (Elite)	10	Habitual carbohydrate intake	N/A	**Objective:** Actigraphy	Increases in daily CHO intake associated with ↓ sleep efficiency (*p* = 0.007) and ↑ WASO (*p* = 0.010)	+
**Protein**											
Leyh et al. [58]	USA	CO (RCT)	10 (0/10)	23.1 ± 1.9	NR (mod-vig activity >4 days/week)	3	Consume either cottage cheese, casein protein, or placebo (no nutrition)	≥2 h after last meal and 30–60 min before sleep	**Objective:** Actigraphy	No significant differences in sleep measures between different protein groups	+
Falkenberg et al. [37]	Australia	PC	36 (36/0)	23.5 ± 3.9	Australian football (Elite)	10	Habitual protein intake and timing	N/A	**Objective:** Actigraphy	Increases in evening (>6 pm) protein intake associated with ↓ sleep latency (*p* = 0.013) Increases in daily protein intake associated with ↓ sleep efficiency (*p* = 0.006), and ↑ WASO (*p* = 0.01)	+
Condo et al. [54]	Australia	PC	32 (0/32)	25.0 ± 4.0	Australian football (Elite)	10	Habitual protein intake	N/A	**Objective:** Actigraphy	No significant association between protein intake and sleep	+
Ferguson et al. [59]	Australia	CO (RCT)	15 (15/0)	22.2 ± 3.6	Australian football (Elite)	4 (2 training and 2 non-training)	55 g whey protein or isocaloric placebo supplement (consumed on 1 × training and non-training day)	3 h pre-bed (≥30 min after dinner)	**Objective:** Actigraphy	No significant difference in all sleep measures following whey protein supplementation	+
Oikawa et al. [60]	Canada	CO (RCT)	11 (5/6)	24.0 ± 4.0	NR (Endurance-trained)	6	20 g α-lactalbumin or collagen after exercise + 40 g before sleep	Post-morning exercise + 2 h pre-bed	**Objective:** Actigraphy	No significant difference in all sleep measures following α-lactalbumin supplementation	+
MacInnis et al. [43]	Canada	CO (RCT)	Study 1—6 (6/0) Study 2—6 (5/1)	Study 1—23.0 ± 6.0 Study 2—24.0 ± 5.0	Cyclists (≥well-trained)	6	Study 1—40 g α-lactalbumin or collagen (×3 nights) Study 2—40 g α-lactalbumin or collagen on night 3 and 6	2 h pre-bed	**Objective:** Actigraphy	No significant difference in all sleep measures following α-lactalbumin supplementation	ø
Miles et al. [61]	Australia	CO (RCT)	16 (0/16)	27.0 ± 7.0	Multiple (trained)	6	40 g α-lactalbumin or 40 g whey (PLA) or 400 mL water (CON)	≥2 h pre-bed	**Objective:** PSG	α-lactalbumin supplementation following simulated evening competition ↑ N-REM 2 % (*p* < 0.05)	ø
**Fat**											
Falkenberg et al. [37]	Australia	PC	36 (36/0)	23.5 ± 3.9	Australian football (Elite)	10	Habitual dietary fat intake and timing	N/A	**Objective:** Actigraphy	No significant association between fat intake and sleep	+
Condo et al. [54]	Australia	PC	32 (0/32)	25.0 ± 4.0	Australian football (Elite)	10	Habitual dietary fat intake	N/A	**Objective:** Actigraphy	Increases in saturated fat intake associated with ↓ sleep latency (*p* = 0.030)	+
**Micronutrients**											
Condo et al. [54]	Australia	PC	32 (0/32)	25.0 ± 4.0	Australian football (Elite)	10	Habitual micronutrient intake	N/A	**Objective:** Actigraphy	Increases in calcium intake associated with ↓ sleep latency (*p* = 0.015) Increases in iron intake associated with ↑TST (*p* < 0.001) and sleep efficiency (*p* < 0.001) Increases in magnesium intake associated with ↓ sleep latency (*p* = 0.031) Increases in sodium intake associated with ↓ TST (*p* < 0.001) Increases in vitamin B12 intake associated with ↑ sleep efficiency (*p* = 0.033), and ↓ WASO (*p* = 0.020) Increases in vitamin E intake associated with ↓ sleep efficiency (*p* = 0.016) Increases in zinc intake associated with ↑ sleep efficiency (*p* = 0.006)	+
**Energy**											
Silva and Paiva [62]	Portugal	Survey (CS)	67 (0/67)	18.7 ± 2.9	Rhythmic gymnastics (Elite)	N/A	Energy intake (<2000 kCal/day)	N/A	**Subjective:** PSQI, ESS	No significant influence of energy intake on sleep	+
Daniel et al. [40]	Brazil	CO (RCT)	9 (9/0)	18.0 ± 0.7	Basketball (State-level)	2	Consume either a high GI dinner and evening snack, or a low GI dinner and evening snack	Dinner + evening snack timing NR	**Objective:** Actigraphy	Increased energy intake correlated with ↓ TST (p NR) and sleep efficiency (*p* < 0.05), and ↑ WASO (*p* < 0.05)	+
Falkenberg et al. [37]	Australia	PC	36 (36/0)	23.5 ± 3.9	Australian football (Elite)	10	Habitual energy and macronutrients	N/A	**Objective:** Actigraphy	Increases in daily energy intake associated with ↑ WASO (*p* = 0.032) Increases in evening energy intake associated with ↑ sleep latency (*p* = 0.011)	+
Condo et al. [54]	Australia	PC	32 (0/32)	25.0 ± 4.0	Australian football (Elite)	10	Habitual energy, macronutrients, and micronutrients	N/A	**Objective:** Actigraphy	No significant influence of energy intake on sleep	+

Abbreviations: CHO (carbohydrates); CO (cross-over); CS (cross-sectional); ESS (Epworth Sleepiness Scale); GI (glycemic index); kCal (Kilocalories); mod-vig (moderate-vigorous); N/A (not applicable); NR (not reported); N-REM 2 (non-rapid eye movement stage 2); PC (prospective cohort); PSG (polysomnography); PSQI (Pittsburgh Sleep Quality Index); RCT (randomized control trial); TST (total sleep time); WASO (wake after sleep onset). ↑ = increase; ↓ = decrease. Quality symbols indicate a positive (+), neutral (ø), or negative (−) study rating.

**Table 3 nutrients-14-03271-t003:** Studies investigating the influence of *dietary supplements* on the sleep of athletically trained populations.

Author (year)	Country	Study Type	Sample Size (m/f)	Age (y)	Sport (Training Status)	Days of Sleep Measurement	Dietary Intervention/Factor		Sleep Tool(s)	Main Outcomes	Study Quality
							Dietary Factor(s)	Timing			
**Caffeine**											
Miller et al. [63]	Australia	CO (RCT)	6 (6/0)	27.5 ± 6.9	Cyclists/triathletes (Well-trained)	2	6 mg/kg caffeine or placebo (2 × 3 mg/kg doses)	3 mg/kg 1 h pre-training (15:50 ± 38 min) + 3 mg/kg 40 min into training (17:40 ± 37 min)	**Objective:** PSG	Caffeine supplementation ↓ TST (*p* = 0.028) and sleep efficiency (*p* = 0.028), and ↑ WASO (*p* = 0.046) compared to placebo	+
Dunican et al. [64]	Australia	PC	20 (20/0)	26.0 ± 3.0	Super Rugby (Professional)	7	Habitual game day caffeine (Mean intake = 2.37 mg/kg)	49 ± 61 min pre-match (match time 19:00–21:00 h)	**Objective:** Actigraphy	Caffeine supplementation ↓ TST (*p* = 0.06) and sleep efficiency (*p* = 0.03), and ↑ sleep latency (*p* = 0.03) compared to placebo	+
Ramos-Campo et al. [65]	Spain	CO (RCT)	15 (15/0)	23.7 ± 8.2	Runners (International and national level)	4	6 mg/kg caffeine or placebo	1 h pre-exercise (18:00)	**Objective:** Actigraphy	Caffeine supplementation ↓ sleep efficiency (*p* = 0.003), and ↑ WASO (*p* = 0.001) and awakenings (*p* = 0.005) compared to placebo	+
Caia et al. [66]	Australia	PC	15 (15/0)	23.0 ± 3.6	Rugby League (Professional)	3	Habitual game day caffeine	*Ad libitum* prior to and during match (match time 19:50)	**Objective:** Actigraphy	Caffeine supplementation on the night of competition ↓ TST (*p* < 0.05) and ↑ sleep latency (*p* < 0.05)	+
Vandenbogaerde and Hopkins [67]	New Zealand	CO (RCT)	9 (6/3)	21–26 *	Swimming (International level)	2	5 mg/kg caffeine or placebo	75 min pre-trial, either morning (09:00–11:30) or evening (17:00–20:00)	**Subjective:** Sleep Quality + Questionnaire	Caffeine supplementation ↓ subjective TST (p NR) and ↑ sleep latency (p NR)	ø
Ali et al. [68]	New Zealand	CO (RCT)	10 (0/10)	24.0 ± 4.0	Team-sports (Recreational to international)	2	6 mg/kg caffeine or placebo	45 min pre-exercise (17:15)	**Subjective:** Leeds Sleep Evaluation Questionnaire	Caffeine supplementation ↑ subjective sleep latency and ↓ sleep quality compared to placebo and baseline (*p* < 0.05)	ø
Raya-Gonzalez et al. [69]	Spain	CO (RCT)	14 (14/0)	21.0 ± 2.0	Basketball (Professional)	2	6 mg/kg caffeine or placebo	60 min pre-fitness testing (18:30–20:00)	**Subjective:** Sleep Quality Questionnaire	Caffeine supplementation ↑ prevalence of insomnia compared to placebo (*p* < 0.05)	ø
Moss et al. [84]	USA	Survey (CS)	234 (104/121) 9 NR	39.5 ± 14.1	Multiple endurance-based sports (NR)	N/A	Usual intake of caffeinated beverages (<1, 1–1.5, >1.5–2, >2–2.5 and >2.5 cups/d)	N/A	**Subjective:** ASSQ	Consuming ≤1.5 cups of caffeinated beverages per day associated with ↑ sleep quality and ↓ sleep difficulty (*p* < 0.05)	+
**Cherry Juice**											
Morehen et al. [70]	United Kingdom	CO (RCT)	11 (11/0)	18.0 ± 1.0	Rugby League (Professional)	6 (24 h pre-match, and 24 and 48 h post-match)	60 mL Montmorency cherry juice or placebo for 5 days pre-match, match day, and 2 days post-match)	2 × 30 mL doses One in morning + one in the evening	**Subjective:** Sleep quality 1–5 scale	No significant difference in sleep quality following Montmorency cherry juice supplementation	+
Wangdi et al. [71]	Australia	Survey (CS)	80 (51/27) 2 NR	27.6 ± 9.8	Multiple sports (≥sub-elite)	N/A	Tart Cherry Juice—supplementation prevalence, and effectiveness	N/A	**Subjective:** General questionnaire	23% of players have previously used or are currently supplementing tart cherry juice, ↑ sleep reported in 14% of those currently or previously taking tart cherry juice	ø
**Pre and Probiotics**											
Harnett et al. [72]	Australia	RCT	19 (19/0)	27.0 ± 3.2	Rugby Union (Elite)	119 (56 domestic, 63 international)	Placebo or 2 × daily Ultrabiotic 60™ + 2 × daily SBFloractiv™ probiotic during international travel	NR	**Subjective:** Sleep quality 1–5 scale	↑ sleep quality following probiotic supplementation (*p* < 0.05)	ø
Quero et al. [73]	Spain	RCT	27 (27/0) Soccer—(14/0) Sedentary—(13/0)	Soccer Placebo— 21.9 ± 2.8 Synbiotic—20.7 ± 1.4	Soccer (Professional) + Sedentary	30	1 × Synbiotic Gasteel Plus^®^ (300 mg) or placebo daily	NR	**Objective:** Actigraphy	In soccer players, Synbiotic^®^ supplementation ↑ sleep efficiency and ↓ sleep latency pre-post intervention (*p* < 0.05)	+
**Other Dietary Supplements**											
Shamloo et al. [74]	Iran	RCT (3-arm)	30 (30/0)	20.7 ± 3.7	NR (‘athletes’)	2 (pre and post supplement)	Consume no drink, placebo, or 100 mL beetroot juice (300 mg nitrates) × 7 days	2 h pre-exercise (exercise timing NR)	**Subjective:** PSQI	Sleep quality ↑ (*p* = 0.001) following beetroot juice	−
Black et al. [75]	New Zealand	RCT	20 (20/0)	22.6 ± 2.9	Rugby Union (Professional)	35	2 × 200 mL protein shakes per day Intervention group + omega-3 (1546 mg)	Post-morning and afternoon exercise	**Subjective:** Sleep quality 1–5 scale	No significant difference in sleep quality between omega-3 and control group	ø
Ormsbee et al. [76]	USA	CO (RCT)	12 (0/12)	29.8 ± 6.5	Runners/triathletes (trained)	2	Placebo or chocolate milk	≥2 h after last meal and <30 min pre-bed	**Subjective:** Self-reported sleep time and normalcy (typical or atypical)	↑ incidence of abnormal sleep following chocolate milk consumption (p NR)	ø
Kasper et al. [77]	United Kingdom	Survey (CS)	517 (517/0)	25.0 ± 5.0	Rugby Union and League (Professional)	N/A	CBD supplementation prevalence, effectiveness, and reasons for trialling the supplement	N/A	**Subjective:** General questionnaire	28% of players aware of CBD were currently or had previously used CBD, 78% of users trialled CBD to improve sleep, ↑ sleep reported in 41% of those currently or previously taking CBD	ø

Abbreviations: CBD (Cannabidiol); CO (cross-over); CON (control); CS (cross-sectional); GI (glycemic index); mod-vig (moderate-vigorous); N/A (not applicable); NR (not reported); PC (prospective cohort); PLA (placebo); PSG (polysomnography); PSQI (Pittsburgh Sleep Quality Index); RCT (randomized control trial); TST (total sleep time); WASO (wake after sleep onset). ↑ = increase; ↓ = decrease. Quality symbols indicate a positive (+), neutral (ø), or negative (−) study rating. * Mean ± SD not available and is presented as a range.

**Table 4 nutrients-14-03271-t004:** Studies investigating the influence of *dietary patterns* on the sleep of athletically trained populations.

Author (year)	Country	Study Type	Sample Size (m/f)	Age (y)	Sport (Training Status)	Days of Sleep Measurement	Dietary Intervention/Factor		Sleep Tool(s)	Main Outcomes	Study Quality
							Dietary Factor(s)	Timing			
**Meal Timing and Patterns**											
Monma et al. [78]	Japan	Survey (CS)	906 (635/271)	19.1 ± 0.8	Multiple sports (“student athletes”)	N/A	Regular mealtimes, skipping breakfast, skipping lunch, skipping dinner, taking meals before bed, taking caffeinated drinks before bed, taking supplements before bed	N/A	**Subjective:** PSQI	No significant influence of dietary factors on sleep quality when adjusted for age, gender, and BMI	ø
Monma et al. [79]	Japan	Survey (CS)	81 (59/22)	32.5 ± 12.0	Multiple Paralympic sports (>50% at national level)	N/A	Regular mealtimes, skipping breakfast, skipping lunch, skipping dinner, taking meals before bed, taking caffeinated drinks before bed, taking supplements before bed	N/A	**Subjective:** PSQI	No significant influence of dietary factors on sleep quality when adjusted for participant attributes	+
Knufinke et al. [80]	Netherlands	Survey (CS)	98 (32/56)	18.8 ± 3.0	Multiple sports (≥national level youth)	N/A	Caffeine consumed after 18:00, Heavy meal within 3 h of bed	N/A	**Subjective:** PSQI, HSDQ, KSS, GSQS, CSD-E	Heavy meal within 3 h of bed associated with ↑TST and an ↑WASO (*p* < 0.05)	+
Hoshikawa et al. [81]	Japan	Survey (CS)	891 (449/368)	>20 *	Multiple sports (Asian Games candidates)	N/A	Eating breakfast every morning	N/A	**Subjective:** PSQI, ESS, Sleep Hygiene Modified Checklist, general questionnaire	Poor sleep quality associated with skipping breakfast (*p* < 0.01)	ø
Falkenberg et al. [37]	Australia	PC	36 (36/0)	23.0 ± 3.9	Australian football (Elite)	10	Habitual meal timing	N/A	**Objective:** Actigraphy	Increases in evening protein intake associated with ↓ sleep latency (*p* = 0.013) Additional hours between main evening meal and bedtime ↓ TST (*p* = 0.042) and WASO (*p* = 0.015)	+
Tinsley et al. [85]	USA	RCT (3-arm)	24 (0/24)	Control 22.0 ± 9.0 TRF 22.1 ± 7.6 TRF_HMB_ 22.3 ± 12.3	NR (resistance training 2–4 days/week)	3 (pre-intervention, 4-week midpoint, post-intervention)	Control diet OR time-restricted feeding OR time-restricted feeding with 3 g/d β-hydroxy β-methylbutyrate supplementation × 8 weeks TRF all calories consumed between 12:00 h and 20:00 h, whereas the control diet was consumed at self-selected intervals	N/A	**Subjective:** PSQI	No changes in PSQI global score within each group or between groups	+
**Total Diet**											
Hoshino et al. [82]	Japan	Survey (CS)	112 (0/112)	19.8 ± 1.0	Multiple sports (college; national level)	N/A	Food Frequency Questionnaire	N/A	**Subjective:** PSQI	No significant difference in nutrient intake between athletes that had a PSQI global score <5.5 or ≥5.5 Greater beans intake for athletes at risk of sleep disorder (PSQI > 5.5) (*p* = 0.034)	+
Moss et al. [84]	USA	Survey (CS)	234 (104/121) 9 NR	39.5 ± 14.1	Multiple endurance-based sports (NR)	N/A	Usual intake of fruit (<1, 1–2, 3–4, 5–6, 7–8, and >8 serves/d), vegetables (<1, 1–2, 3–4, 5–6, 7–8, and >8 serves/d), wholegrains (<1, 1–2, 3–4, 5–6, 7–8, 9–10, 11–12, and >12 serves/d)	N/A	**Subjective:** ASSQ	No significant influence of fruit, vegetable, or wholegrain intake on sleep difficulty or sleep quality	+
**Dairy Consumption**											
Yasuda et al. [83]	Japan	Survey (CS)	679 (379/300)	25.1–26.0 *	Multiple sports (Olympic games candidates)	N/A	Frequency of milk or dairy consumption (d/wk)	N/A	**Subjective:** Sleep quality (1–3 scale), general questionnaire	Higher milk consumption associated with ↓ risk of poor sleep quality in female athletes only (*p* < 0.001)	ø
Moss et al. [84]	USA	Survey (CS)	234 (104/121) 9 NR	39.5 ± 14.1	Multiple endurance-based sports (NR)	N/A	Usual intake of dairy milk (<1, 1–2, 3–4, 5–6, 7–8, and >8 cups/d)	N/A	**Subjective:** ASSQ	No significant influence of dairy milk intake on sleep difficulty or sleep quality	+

Abbreviations: ASSQ (Athlete Sleep Screening Questionnaire); CS (cross-sectional); CSD-E (Consensus Sleep Diary Expanded); ESS (Epworth Sleepiness Scale); GSQS (Groningen Sleep Quality Scale); HSDQ (Holland Sleep Difficulty Questionnaire); KSS (Karolinska Sleepiness Scale); PSQI (Pittsburgh Sleep Quality Index); TRF (time-restricted feeding); TRF_HMB_ (time-restricted feeding with β-hydroxy β-methylbutyrate supplement). ↑ = increase; ↓ = decrease. Quality symbols indicate a positive (+), neutral (ø), or negative (−) study rating. * Mean ± SD not available and is presented as a range.

## Data Availability

Not applicable.

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
