# Peer review of "The Impact of Dietary Factors on the Sleep of Athletically Trained Populations: A Systematic Review"

_nutrients, 2022, doi:10.3390/nu14163271_

Round 1

Reviewer 1 Report

This review discussed the influence of dietary intakes on sleep metrics and overall, was a well-written and insightful work. The authors clearly state the methodological approaches and interpret the literature cautiously. I have only minor suggestions which are mostly the inclusion of literature to improve the argument of the manuscript and the interpretation of the results.

Line 36: Do athletes require additional sleep (relative to non-athletes) or additional sleep + higher sleep quality (assuming amount of sleep is not the only factor)? I would clarify here to make the argument stronger.

Line 37-38: Add the citation PMID: 35225908 discussing recreational athletes. It will fit well in the sentence given that their study found that 93% of recreational endurance athletes believe in the benefits of sleep (then transition into the issues with elite athletes; this is only a suggestion).

Lines 41 through 46: I agree with lines 41 through 44, but I imagine that athletes who do not prioritize adequate sleep also do not consider other important aspects of training (lines 45-46), and thus, sleep is a facet of the higher likelihood to lose in competition as suggested in lines 41-44. I would suggest ending line 43 with “all impaired, all of which may diminish sporting success” since you really describe a number of things that may contribute to losing in that sentence anyways.

Lines 57-58: Clarify that this would be for elite athletes, or suggest who would be subjected to these anti-doping guidelines. PMID: 35380079 found that melatonin was one of the most used supplements in a group of endurance athletes, specifically for older athletes (> 40 years). So, I assume this sentence would not pertain to high-level competitive recreational athletes.

Line 59-61: Great work here.

Line 70-72: You suggest that the athlete's schedule is complex, so would it be reasonable to suggest that their dietary intakes and meal timing often need to be in the evening hours (due to scheduling) to meet certain thresholds? Thus, what should be consumed and how many hours before bedtime they should be consuming it should be suggested.  I see in lines 73-85 that you elaborate on meal composition. Could you elaborate on the sources of protein that have high TRP:LNAAs which would improve sleep? Further, and you show this later, the effect of pre-sleep protein for performance (not sleep) is clear (PMID: 32811763, 27916799, 30895177) so I would consider the “trade-off” when suggesting alterations in pre-sleep protein consumption.

Line 90-91: Adding PMID: 35308592 to this sentence would improve this considering that they found whole grains to be associated with lower sleep issues in endurance athletes.

Line 137: Could you elaborate on this? Specifically, is the 3x/wk for 5h/wk threshold specific to training for a sport, or would this also consider active exercisers who would go to the gym for an hour a day? This would be helpful.

Line 142-157: Great work.

Line 196: PMID: 35308592 was a cross-sectional survey design that meets the criteria and was published in March 2022. I would consider adding this manuscript, especially considering their findings on caffeinated beverages that support your study conclusions.

Line 436: Although they may not fit within your study limitations, the Meal Timing and Patterns section would improve from the inclusion of studies examining time-restricted feeding (intermittent fasting and Ramadan) in athletes. Some studies to consider adding: PMID: 27550719, 27737674, 31268131, 29910364, 24524697, 34198990

Line 459: PMID: 35308592 discusses the relationship between milk and sleep in endurance athletes. Consider its inclusion.

Line 510-512: I know that your role is to report the findings, but I would consider adding a sentence after this sentence discussing a “practical approach” as you would certainly not want to suggest that total protein intake should be reduced to improve sleep as I do not believe that either would be a beneficial trade-off. Consider future research on a strategy that allows for high-adequate protein + quality or unaffected sleep (concomitant reductions in caffeine or probiotic inclusion, perhaps?).

Line 531: Typo, add “glycemic” after “high”

Line 560: Great take (and how I think the real-world approach should be added for protein consumption. Caffeine consumed for training (rather than competition only) would be nice to include given that some athletes (ex: non-professional athletes who would need to train after day jobs) might consume caffeine prior to any athletic endeavor (training and competition) and how daily caffeine supplementation should be altered or scheduled to lessen the effect on sleep.

Line 584: Consider adding PMID: 29364545 suggesting that exercise is beneficial for the gut microbiota but worsened with excessive exercise (a potential issue for athletes).

Line 613: PMID: 32757826 suggests that >90% of people found CBD to be somewhat or very helpful. Consider including it to suggest more research on athletes with sleep problems.

Line 644: Personally, I would not add probiotics to the “practical applications” section as there is too much unknown. Specifically, the initial microbiome is a significant determinant of the probiotic’s effect (what if you have a “healthy” microbiome and you supplement a single strain in high doses, ultimately lowering the diversity), the strains within the probiotic, the variety of the person's diet, and the exercise regimens. If you choose to keep this section, I suggest stating that expectations for improving sleep with probiotic supplements should consider these or “certain” factors. 

Author Response

Thank you for taking the time to provide comments that will strengthen this systematic review. Please see the attached .pdf file for responses to the comments provided.

Author Response

Thank you for putting the time to provide detailed comments that have helped to make this a higher quality manuscript. Please see the attached .pdf file for responses to the comments provided.
